# Normalizing Flows for High-Dimensional Detector Simulations

Florian Ernst[1,2], Luigi Favaro[1], Claudius Krause[1,3], Tilman Plehn[1], and David Shih[4]

**1** Institut für Theoretische Physik, Universität Heidelberg, Germany
**2** Experimental Physics Department, CERN, Geneva, Switzerland
**3** Institut für Hochenergiephysik (HEPHY), Österreichische Akademie der Wissenschaften (ÖAW), Vienna, Austria
**4** NHETC, Department of Physics & Astronomy, Rutgers University, Piscataway, NJ USA

December 22, 2023

## Abstract

Whenever invertible generative networks are needed for LHC physics, normalizing flows show excellent performance. A challenge is their scaling to high-dimensional phase spaces. We investigate their performance for fast calorimeter shower simulations with increasing phase space dimension. In addition to the standard architecture we also employ a VAE to compress the dimensionality. Our study provides benchmarks for invertible networks applied to the CaloChallenge.

# 1  Introduction

Simulations are a defining aspect of LHC physics, bridging experiment and fundamental theory and allowing for a proper interpretation of LHC measurements [1, 2]. The simulation chain starts from the hard interaction at the scattering vertex, and progresses through the radiation of soft particles, the decay of heavy, unstable particles, hadronization of colored states, and subsequent interaction of all particles in the event with the detector. In the development of LHC as a precision-hadron collider, the last step has become a major bottleneck in speed and precision, in particular the reproduction of the detailed interactions of incident and secondary particles within the calorimeters. Generating these calorimeter showers with GEANT4 [3–5], based on first principles, takes a substantial amount of the LHC computing budget. Without significant progress, simulations will be the limiting factor for all analyses at the high-luminosity upgrade of the LHC.

One development driving faster LHC simulations is the advent of deep generative networks. Such networks have shown great promise for LHC physics in the past few years, providing fast and accurate surrogates for simulations in high-dimensional phase spaces [6]. They learn the underlying probability distribution of events or calorimeter showers from a reference dataset and then generate new samples based on this learned distribution [7,8]. We have seen successful applications to all steps in the simulation chain [2], phase space integration [9–19]; parton showers [20–27]; hadronization [28–31]; detector simulations [32–61]; and end-to-end event generation [62–68].

For LHC physics, it is crucial that these networks are not used as black boxes, but their performance can be investigated, understood, and improved systematically [67, 69–73]. This is especially important when their conditional counterparts are used for inference [74–76], probabilistic unfolding [77–86], or anomaly detection [87–92].

In this paper, we will focus on the problem of building fast and accurate surrogate models for calorimeter shower simulation, using the technology of normalizing flows. We have seen in a number of contexts that normalizing flows are a promising technique for fast calorimeter simulation [41, 43, 49, 73], but there are also major challenges with scaling them up to more granular (higher-dimensional) calorimeters [50, 51, 56, 61]. These challenges are especially interesting because recently, continuous-time generative models (diffusion models and continuous normalizing flows trained with flow-matching) have entered LHC physics [25–27, 47, 55, 57, 58, 68, 72, 76, 93–95] and show impressive performance which is not as limited by the dimensionality of the data. However, their gain in expressivity comes at the expense of slower generation, leading to an interesting trade-off between speed and quality of generated events or showers.

Here, we will build on previous works [50, 51, 56, 61] attempting to scale up normalizing flows to higher-granularity calorimeters. Focusing on the datasets [96–99] of the Fast Calorimeter Simulation Challenge [100], we will tackle this problem in two ways.

- First, we will show how impressive gains in both speed and accuracy can be achieved by switching from the fully-autoregressive flows of [41, 43, 49, 51, 56] to flows based on coupling layers [101–103] which are equally fast in the sampling and density estimation directions. Following the terminology of [104, 105], we will refer to coupling-layer based flows as *invertible neural networks* (INNs) throughout this work. Using the INN framework, we are able to obtain state-of-the-art results on dataset 1 (pions and photons) and dataset 2 of the CaloChallenge.

- Second, to reach the dimensionality of dataset 3, we will combine the INN framework with a VAE. Similar to the approaches of [50] and [106], we will train the INN on the (much lower-dimensional) latent space of a VAE fit to the showers of dataset 3. Then sampling from the INN and passing this through the decoder of the VAE, we will obtain a surrogate

model for dataset 3. We will see that the results here, while not state-of-the-art in terms of quality, are very fast to generate, so could fill out another point in the Pareto frontier of fast calorimeter shower simulation.

The paper starts by introducing the CaloChallenge datasets in Sec. 2. In Sec. 3 we introduce our fast INN version [104,105] of a normalizing flow, as well as a VAE+INN combination. In Sec. 4 we discuss their performance on the different dataset, with increasing phase space dimensionality and including learned classifier weights. We conclude and provide timing information in Sec. 5. In the Appendices we provide details on the different network architectures and hyperparameters and compare the INN performance to CaloFlow.

## 2 Datasets

We use the public datasets [96–99] of the Fast Calorimeter Simulation Challenge [100]. They consist of showers simulated with GEANT4 for different incident particles. The general geometry is the same across all datasets: the detector volume is segmented into layers in the direction of the incoming particle. Each layer is segmented along polar coordinates in radial ($r$) and angular ($\alpha$) bins. A shower is given as the incident energy of the incoming particle and the energy depositions in each voxel.

Dataset 1 (DS1) provides calorimeter showers for central photons and charged pions. They have been used in ATLFAST3 [42]. The voxelizations of the 5 photon layers and 7 pion layers in radial and angular bins ($n_r \times n_\alpha$) are

$$
\begin{aligned}
\text{photons} \quad & 8 \times 1,\ 16 \times 10,\ 19 \times 10,\ 5 \times 1,\ 5 \times 1 \\
\text{pions} \quad & 8 \times 1,\ 10 \times 10,\ 10 \times 10,\ 5 \times 1,\ 15 \times 10,\ 16 \times 10,\ 10 \times 1
\end{aligned}
\tag{1}
$$

This gives 368 voxels for photons and 533 voxels for pions. The incoming particles are simulated for 15 different incident energies $E_{\text{inc}} = 256$ MeV ... 4.2 TeV, increasing by factors of two, with the sample sizes given in Tab. 1. The original ATLAS dataset does not require an energy threshold. The effect of a threshold on the shower distributions at the detector cell level requires further studies. We require $E_{\text{min}} = 1$ MeV to all generated voxels, motivated by the readout threshold of the calorimeter cells and the fact that photon showers require a minimum cell energy of 10 MeV to cluster and pion showers start clustering at 300 MeV [107].

Datasets 2 and 3 (DS2/3) are not modeled after existing detectors. They assume 45 layers of active silicon detector (thickness 0.3 mm), alternating with inactive tungsten absorber layers (thickness 1.4 mm) at $\eta = 0$. Each dataset contains 100,000 GEANT4 positron showers with log-uniform $E_{\text{inc}} = 1$ ... 1000 GeV. The only difference between the two datasets is the voxelization. In dataset 2, each layer is divided into $16 \times 9$ angular and radial voxels, defining 6480 voxels in total. Dataset 3 uses $50 \times 18$ voxels per layer or 40,500 voxels in total. The minimal recorded energy per voxel for these two datasets is 15.15 keV.

| $E_{\text{inc}}$ | 256 MeV ... 131 GeV | 256 GeV | 0.512 GeV | 1.04 TeV | 2.1 TeV | 4.2 TeV |
|---|---|---|---|---|---|---|
| photons | 10000 per energy | 10000 | 5000 | 3000 | 2000 | 1000 |
| pions | 10000 per energy | 9800 | 5000 | 3000 | 2000 | 1000 |

Table 1: Sample sizes for different incident energies in dataset 1.

# 3 CaloINN

We study two different network architectures. First, we benchmark a standard INN and demonstrate its precision and generation speed especially for low-dimensional phase space. Second, we embed this INN in a VAE, with the goal of describing datasets 2 and 3 with the same physics content, but a much larger phase space dimensionality.

## 3.1 INN

Normalizing flows describe bijective mappings between a (Gaussian) latent space $r$ and the physical phase space $x$,

$$
p_{\text{latent}}(r) \quad \underset{\leftarrow \overline{G}_\theta(x)}{\overset{G_\theta(r)\rightarrow}{\longleftrightarrow}} \quad p_{\text{model}}(x) \sim p_{\text{data}}(x) \, . \tag{2}
$$

$\overline{G}_\theta(x)$ denotes the inverse transformation to $G_\theta(r)$. The INN variant [104,105] of normalizing flows is completely symmetric in the two directions. After training the network, $p_{\text{data}}(x) \sim p_{\text{model}}(x)$, we use the INN to sample $p_{\text{model}}(x)$ from $p_{\text{latent}}(r)$ [6].

The building block of our INN architecture is the coupling layer [101–103]. We replace the standard affine layer by a more expressive spline transformation. We use different spline transformations, depending on speed and expressivity. For datasets 1, we employ a rational quadratic spline [108], while for dataset 2 we use a cubic spline [109]. All INN hyperparameters are given in Tab. 3.

The INN is implemented using the FREIA[§] package [110], and is trained with a likelihood loss,

$$
\mathcal{L}_{\text{INN}} = -\left\langle \log p_{\text{model}}(x) \right\rangle_{p_{\text{data}}} = -\left\langle \log p_{\text{latent}}\big(\overline{G}_\theta(x)\big) + \log \left| \frac{\partial \overline{G}_\theta(x)}{\partial x} \right| \right\rangle_{p_{\text{data}}} . \tag{3}
$$

The first term ensures that the latent representation remains Gaussian, while the second term constructs the correct transformation to the phase space distribution. Given the structure of $\overline{G}_\theta(x)$ and the latent distribution $p_{\text{latent}}$, both terms can be computed efficiently.

As a noteworthy preprocessing we normalize each shower to the layer energy. The energy information is encoded as

$$
u_0 = \frac{\sum_i E_i}{E_{\text{inc}}} \qquad \text{and} \qquad u_i = \frac{E_i}{\sum_{j \geq i} E_j} \, , \tag{4}
$$

in terms of the energy depositions per layer $E_i$. The $u_i$ are appended to the list of voxels for each shower. We do not explore a separate training for the energy and the voxel dimensions which would simplify the learning process of the energy dimensions. We train the INN on the full data, conditioned on the logarithm of the incident energies. Unlike, for instance, CaloFlow [43,49,56] we train a single network without any distillation. We provide the details of the preprocessing in App. A.

## 3.2 VAE+INN

The problem with the INN is the scaling towards dataset 3 with its high-dimensional phase space of 40k voxels. To solve this scaling problem we introduce an additional VAE to reduce the dimensionality of the INN mapping. Differently from [50], we do not estimate the dimensionality of the manifold but rather optimize the reconstruction of the VAE while keeping a

---

[§]We provide the code in a Github repository at https://github.com/heidelberg-hepml/CaloINN

low-dimensional latent space. The VAE consists of a preprocessing block, an encoder-decoder combination, and a postprocessing block. Both, the decoder and the encoder are conditioned on the incident energies and additional energy variables. Therefore, we compress normalized showers in the latent space and jointly learn the energy and the latent variables with the INN. During generation, the INN samples into the latent space of the VAE, and the VAE decoder translates this information to the shower phase space. We set the latent space to 50 for dataset 1 and dataset 2, and to 300 for dataset 3. Other specifics of the network are different in the three datasets and are provided in App. A.

The goal of our standard $\beta$VAE is to learn to reconstruct the input data. We assume a Gaussian distribution for the encoder network $E(z|x)$. The VAE loss for the compression is

$$\mathcal{L} = \mathcal{L}_{\text{BCE}} + \beta D_{\text{KL}}[E(z|x), p_{\text{latent}}(z)] \,, \tag{5}$$

with the usual binary cross entropy loss and the Gaussian prior. For a Gaussian encoder the KL-divergence can be computed analytically, and the coupling strength is $\beta = 10^{-9}$.

For the decoder we use a Bernoulli likelihood, because it outperforms for example its Gaussian counterpart. The Gaussian decoder does not model the shower geometry well, and it under-populates the low-energy regions. The continuous Bernoulli distribution [111] leads to instabilities, as the average energy deposition in the normalized space is close to zero. We use a Bernoulli decoder,

$$D(x|\lambda(z)) = \lambda(z)^x (1 - \lambda(z))^{1-x} \,, \tag{6}$$

defining the combined VAE loss

$$\mathcal{L}_{\text{VAE}} = \left\langle \left\langle x \log \lambda + (1-x) \log(1-\lambda) \right\rangle_{z \sim E(z|x)} + \beta \left[ 1 + \log \sigma_E^2 - \mu_E^2 - \sigma_E^2 \right] \right\rangle_{x \sim p_{\text{data}}} \,. \tag{7}$$

Because the Bernoulli distribution gives a binary probability we use its continuous mean $\lambda$ as the prediction for the individual voxels.

The remaining differences between the unit-Gauss prior in the latent space and the encoder are mapped by the INN. Applying a 2-step training we first train the VAE and then train the INN given the learned latent space. This means we pass the encoder means and the standard deviations, as well as the energy variables to the INN. The INN is trained as described above, mapping the latent representation of the VAE to a standard Gaussian. As for the full INN, the energy information is encoded following Eq (4) and learned by the latent flow. Both encoder and decoder of the VAE are conditioned to these variables.

For the larger datasets 2 and 3, we employ a mixture of a convolutional and a fully connected VAE. Our assumption is that the calorimeter layers do not require a full correlation, so we can simplify the structure by compressing consecutive layers jointly in a first-step compression. We use an architecture with fully connected sub-blocks, resembling a kernel architecture with a kernel size $k$ (number of jointly encoded calorimeter layers) and a stride $s$ (distance between two neighboring kernel blocks). After this first compression we concatenate these latent sub-spaces and compress them a second time into our final latent space. For the decoding we reverse this two-step structure. The overlapping regions of the fully connected kernel blocks are summed over.

## 4   Results

The main physics reason for specific shower features is the incident energy. Low-energy showers will interact with only a few layers of the calorimeter and quickly widen, leading to a broad

center-of-energy distribution in earlier calorimeter layers and a high sparsity in the given voxelization. High-energy showers penetrate the calorimeter more deeply. They will be collimated in the initial layers and have low sparsity since each shower is likely to deposit energy in each voxel.

To see if the ML-learned showers reflect these physics properties, we look at physics-motivated and high-level features. Given a shower with energy depositions $\mathcal{I}$, we look at the center of energy and its width for each layer,

$$\langle \zeta \rangle = \frac{\zeta \cdot \mathcal{I}}{\sum_i \mathcal{I}_i} \qquad \text{and} \qquad \sigma_{\langle \zeta \rangle} = \sqrt{\frac{\zeta^2 \cdot \mathcal{I}}{\sum_i \mathcal{I}_i} - \langle \zeta \rangle^2} \qquad \text{for} \qquad \zeta = \eta, \phi \; ; \tag{8}$$

where $\sum_i$ runs over the voxels in one layer. We also look at the energy deposition in each layer; the layer sparsity; and for dataset 1, the ratio $E_{\text{tot}}/E_{\text{inc}}$ for each discrete incident energy.

To analyze the quality of our generative networks in more detail and to identify failure modes, we train a classifier $D(x)$ on the voxels, to distinguish GEANT4 showers from generated showers [41, 43, 73]. By the Neyman-Pearson lemma the trained classifier approximates the likelihood-ratio. This means we can compute the correction weight [73] and use the weight distributions as an evaluation metric

$$w(x) = \frac{D(x)}{1 - D(x)} \approx \frac{p_{\text{data}}}{p_{\text{model}}}(x) \,. \tag{9}$$

For these weights it is crucial that we evaluate them on the training and on the generated datasets combined, because typical failure modes correspond to tails for one of the two datasets [73]. In addition, we always check if showers with especially small or large weights cluster in phase space, allowing us to identify failure modes of the respective generative network.

## 4.1 Dataset 1 photons

We start with the photons in dataset 1, the simplest case in terms of dimensionality and of complexity, since photons only undergo a handful interactions in the calorimeter. In this established benchmark normalizing flows are known to excel. We summarize the most interesting high-level features for the GEANT4 training data, the INN generator, and the VAE+INN generator in Fig. 1.

For instance for the calorimeter layer 2, we first look at the shower shape in rapidity. Dataset 1 is not a symmetric in $\eta$ and $\phi$, because the shower were not generated around $\eta = \phi = 0$. All showers have the same mean width, regardless of the incident energy. This is captured by both networks at the level of 5% to 20%. A failure mode of the INN is the region $\sigma_{\langle \eta \rangle} < 20$ mm. where the network undershoots the training data by up to 30%. A peculiar feature of these distributions is a small peak at zero, which occurs when at most one voxel per layer receives a hit. These cases are better reproduced by the VAE, whereas the INN tends to produce slightly more collimated showers.

The sparsity $\lambda_2$ in the same layer is determined by the energy threshold of 1 MeV. The INN matches the truth over the entire $\lambda$-range to 10%, while the VAE struggles. In particular, its showers have too many active voxels, leading to the mis-modeled peak close to zero.

Next, we show the energy depositions in layers 0 and 2. Both networks show comparable performance over the entire energy range and are challenged by the sharp increase in energy around 100 MeV. The energy in layer-2 highlights the effect of discrete incident energies. High-energy showers at fixed energy deposit a similar amount of energy creating steps in the histogram.

Finally, the ratio $E_{\text{tot}}/E_{\text{inc}}$ exhibits a small bias in the energy generation for the VAE+INN towards low energies, artifact of the final threshold in the architecture. For smaller incident

energies, more voxels are zero [51], which causes a problem for the VAE+INN because we already know that the sparsity is its weakness.

To illustrate the discrete structure of the incident energies in dataset 1, we collect $E_{tot}/E_{inc}$ for each incident energy in Fig. 2. The incident energy, provided during training and generation, carries energy-dependent information about the shower. For instance, low-energy showers have a much broader energy ratio distribution, in contrast to high-energy showers. Both generative networks learn the conditional distribution on $E_{inc}$ with deviations up to 30% in the tails.

## 4.2 Dataset 1 pions

The physics of hadronic showers is significantly more complex than photon showers, so it is interesting to see how our INNs perform for a low-dimensional calorimeter simulation of pions. As before, we show shower shapes, sparsity, energy depositions, and the fraction of deposited energy in Fig. 3.

For the shower shapes, both networks show small, percent-level deviations in the bulk of the distributions. In addition, the VAE+INN is smearing out secondary peaks of the distributions. Both networks generate slightly too wide showers, predominantly where the energy deposition ends up in a narrow band of the calorimeter.

The slightly reduced quality of the generated pion showers can also be seen in the sparsity, especially for the VAE+INN. The energy depositions indicate similar failure modes. The sharp cut at low energy is smeared to a different extent by the networks, and a second deviation appears in the low-density region before the sharp cut. From the ratio $E_{tot}/E_{inc}$ we see that at all energies the fraction of deposited energy can be very different from shower to shower, leading to the wide energy distribution far from one.

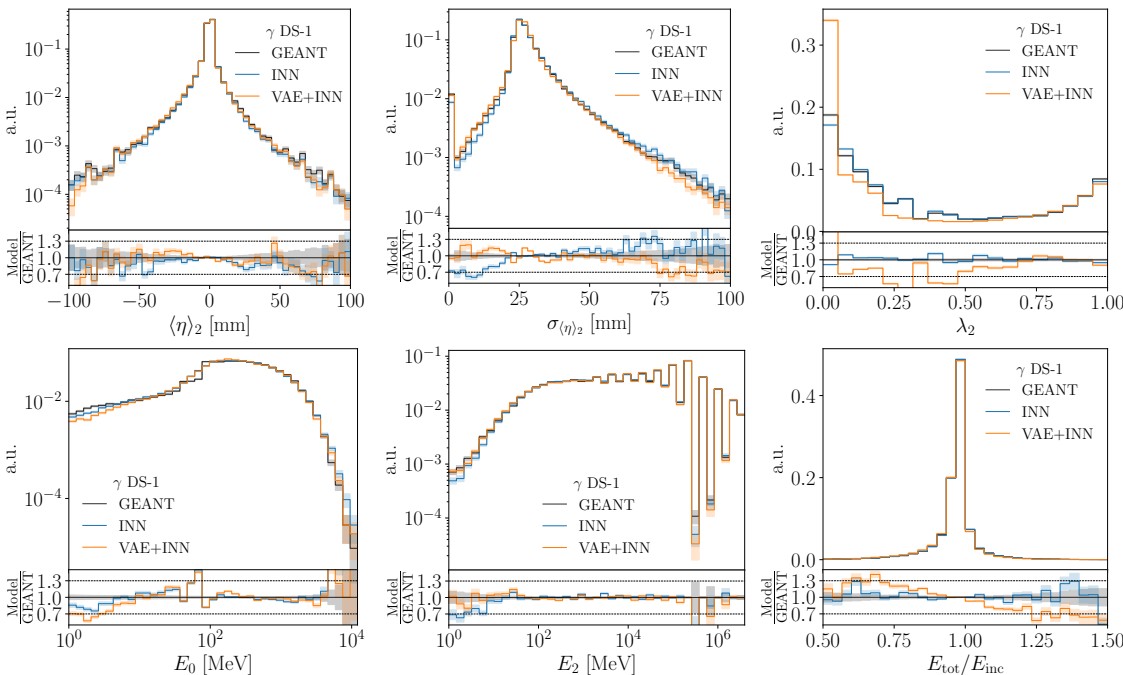

Figure 1: Set of high-level features for $\gamma$ showers in dataset 1, compared between GEANT4, INN, and VAE+INN.

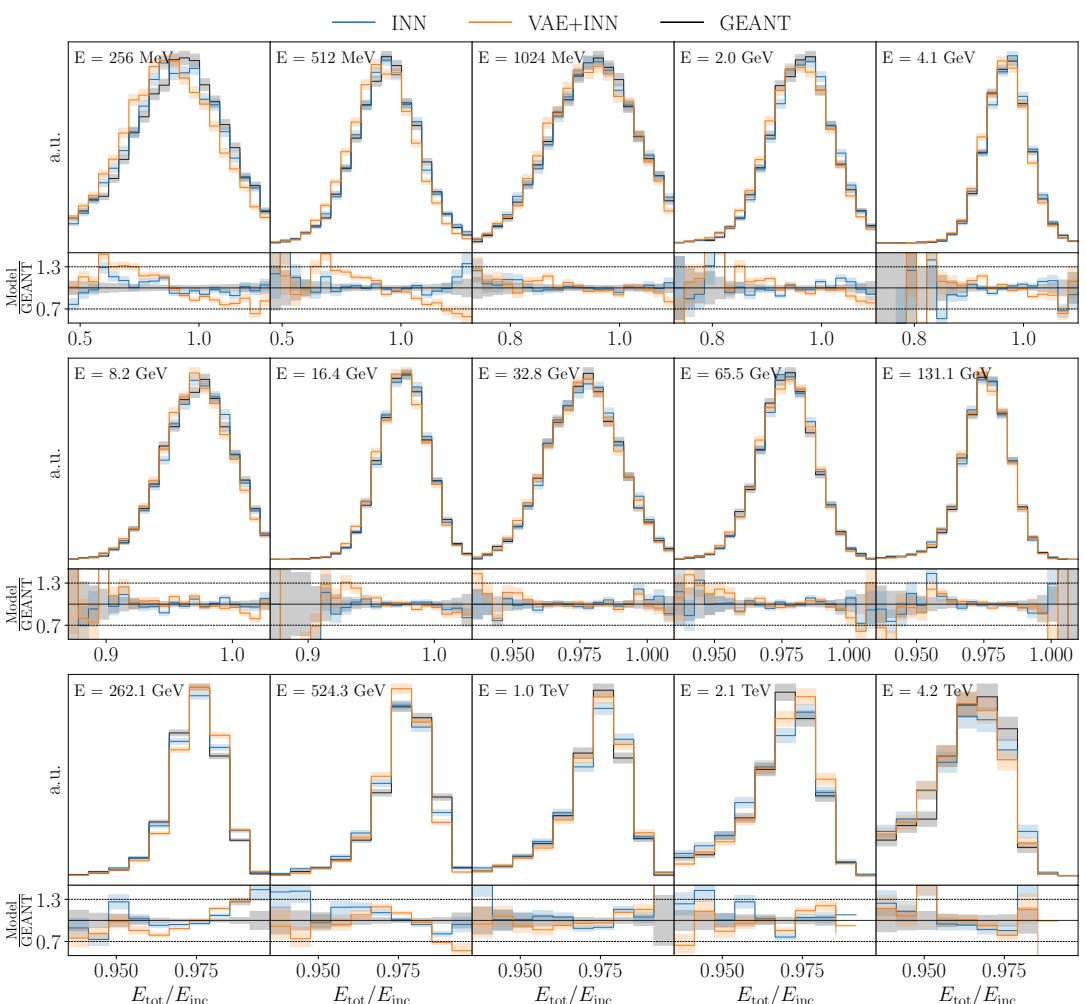

Figure 2: Energy ratio $E_{tot}/E_{inc}$ for each discrete incident energy, compared between GEANT4, INN, and VAE+INN for $\gamma$ showers.

**Low-level classifier**

To evaluate the performance of our generative networks on dataset 1 systematically, we train a network to learn the classifier weights defined in Eq.(9) over the voxel space. In the left panel of Fig. 4 we show the weights for the $\gamma$-shower. We clearly see that the INN outperforms the VAE+INN. Its weight distribution peaks much closer to 1 and the corresponding AUC of 0.601 is substantially better than the corresponding AUC=0.936 of the VAE+INN.

More importantly, the INN does not show significant tails at large or small weights, which would indicate distinct failure modes. The peak of the VAE+INN, on the other hand, has moved away from 1. The tail at small weights indicates regions that are overpopulated by the network. We already know that this is the case for the sparsity. Large weights appear in phase space regions which the VAE+INN fails to populate, for instance the widths of the centers of energy.

In the right panel of Fig. 4 we see that the two generators perform more similar for $\pi$-showers. Both networks now show tails at small and large weights, two orders of magnitude away from one. This means there are regions that are over- and underpopulated by the generative networks. The fact that small weights appear for generated showers and large weights appear for the training data is generally expected. The AUCs for the INN and the VAE+INN are

0.805 and 0.864, respectively. The INN weight distribution is sharper around one, resulting in the smaller AUC compared to the combined VAE+INN approach. Altogether, we find that for dataset 1 with its limited dimensionality of 368 voxels for photons and 533 voxels for pions the INN works well, and that adding a VAE to compress the information does not help with the network performance.

### 4.3   Dataset 2 positrons

Dataset 2 is given in terms of 6480 voxels, the kind of dimensionality which will probe the limitations of the regular INN. The number of parameters for this network approaches 200M. The question will be, if the VAE+INN condensation helps the performance of the network. As before, we show a representative set of high-level features in Fig. 5. We choose layer 20, approximately in the middle of the calorimeter. It combines features from low-energy showers, which are absorbed in this region, and high-energy showers, which continue until the end of the calorimeter.

From the shower shapes we see that the INN-based architectures generate realistic showers at all energies. The training is stable and consistent across different runs of the same architecture. We only see sizeable deviations in the center of energy distributions in the first and last layers of the calorimeter, where there are less energy depositions. The agreement in phase space density between GEANT4 and the INN ranges from a few percent in the bulk of the distributions to 50% in the tails. Similar numbers apply to the width of the center of energy. The failure mode of the INN, regardless of the dataset, is an under-sampling of showers with width between the peak at zero and the secondary peak, for which the location depends on the layer but not on the incident energy. The VAE+INN generates showers of slightly worse quality. This is true for the shower shapes, for instance the peak position in $\sigma_{\langle\eta\rangle}$, but most obvious for the poorly reproduced sparsity.

The two networks learn the energy depositions in the layers in two very different spaces. The INN extracts them with a large number of voxels, while the VAE+INN compresses them

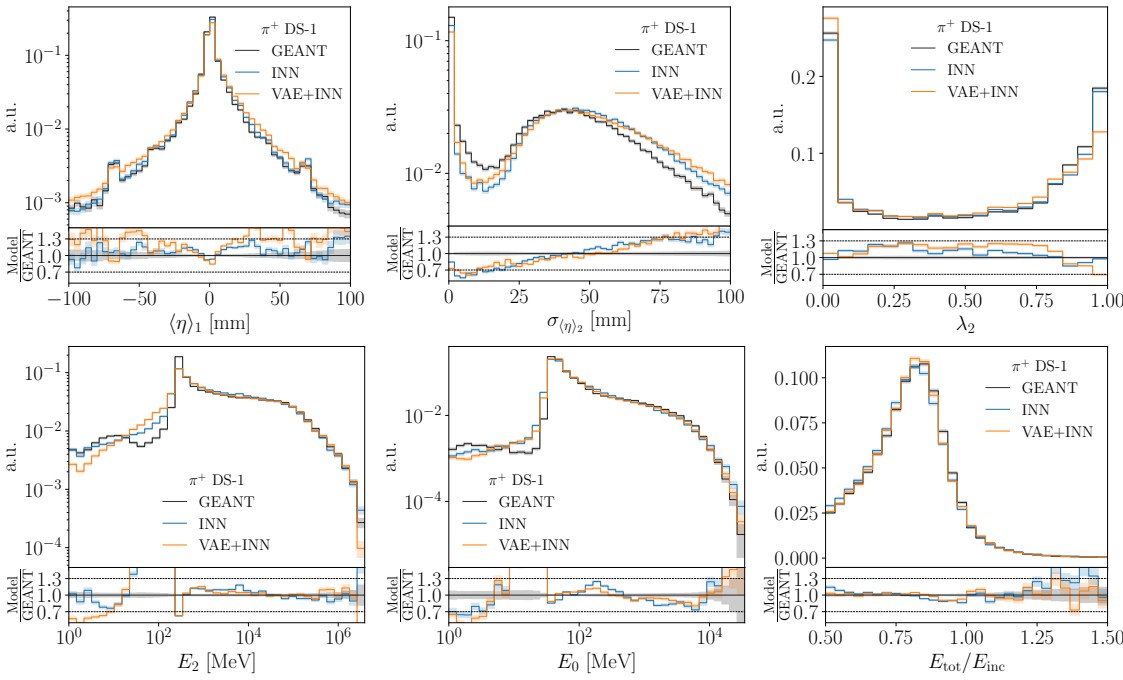

Figure 3: Set of high-level features for pion showers in dataset 1, compared between GEANT4, INN, and VAE+INN.

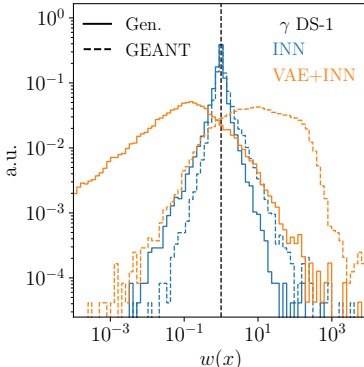 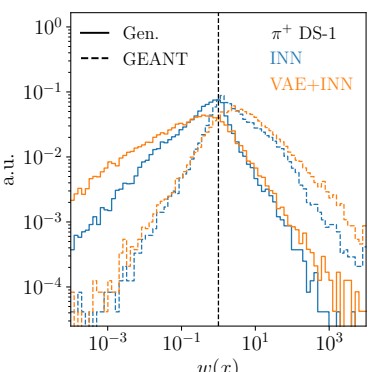

Figure 4: Classifier weight distributions in dataset 1. Classifier trained on $\gamma$ showers (left) and $\pi$ showers (right).

into a reduced space of around 50 additional features. This different expressivity is reflected in all energy distributions in Fig. 5. While in poorly populated tails the INN does better, for instance at low energies, the VAE+INN performs better for the main features in the central and high-energy regime. This is true for the layer-wise energies, but also for the ratio $E_{tot}/E_{inc}$.

**Low-level classifier**

Again, we show a systematic comparison for dataset 2 in terms of the classifier weights in the left panel Fig. 6. Compared to dataset 1, there is a clear deterioration of the INN performance for the higher-dimensional phase space. At small weights, the tail remains narrow, indicating that there are still no phase space regions where the network over-samples the true phase space distribution. For large weights the weight tail now extends to values larger than $w \sim 10^3$. This tail can be related to a recurrent under-sampling of showers with a small width of the center of energy in each layer, as seen in Fig. 5.

The classifier evaluating the VAE+INN generator highlights a few important structures as well. First, we have a clear over-sampled region in phase space with weights $w \sim 10^{-2}$, which we can relate to the center of energy distribution as well. As mentioned before, the VAE+INN over-samples showers with width close to the mean shower width. The classifier confirms this major failure mode. For the large-weight tail we checked that the under-sampled showers do not cluster in the same way, but are distributed over phase space, including tails of distributions.

The AUC values of the classifiers for dataset 2, 0.703 for the INN and 0.916 for the VAE+INN, confirm the challenge of the INN, especially relative to the well-modelled $\gamma$-showers in dataset 1. However, adding a VAE does not significantly improve the situation as long as it is technically possible to train an INN.

## 4.4 Dataset 3 positrons

Finally, we tackle dataset 3, which includes the same physics as dataset 2, but over a much higher-dimensional and extremely sparsely populated phase space. For this dataset we cannot train an INN without dimensionality reduction, so we only show VAE+INN results in Fig. 7. As expected, the performance is worse than for dataset 2, but the training is stable across different training runs. One problem is a worsening reconstructions of the centroids and widths for the later layers, which is likely related to the small average energy deposition per voxel. The maximum in the width distributions is overpopulated by the VAE+INN. For the energy distributions the VAE+INN is doing reasonably well, with serious deviations only in the low-energy tails.

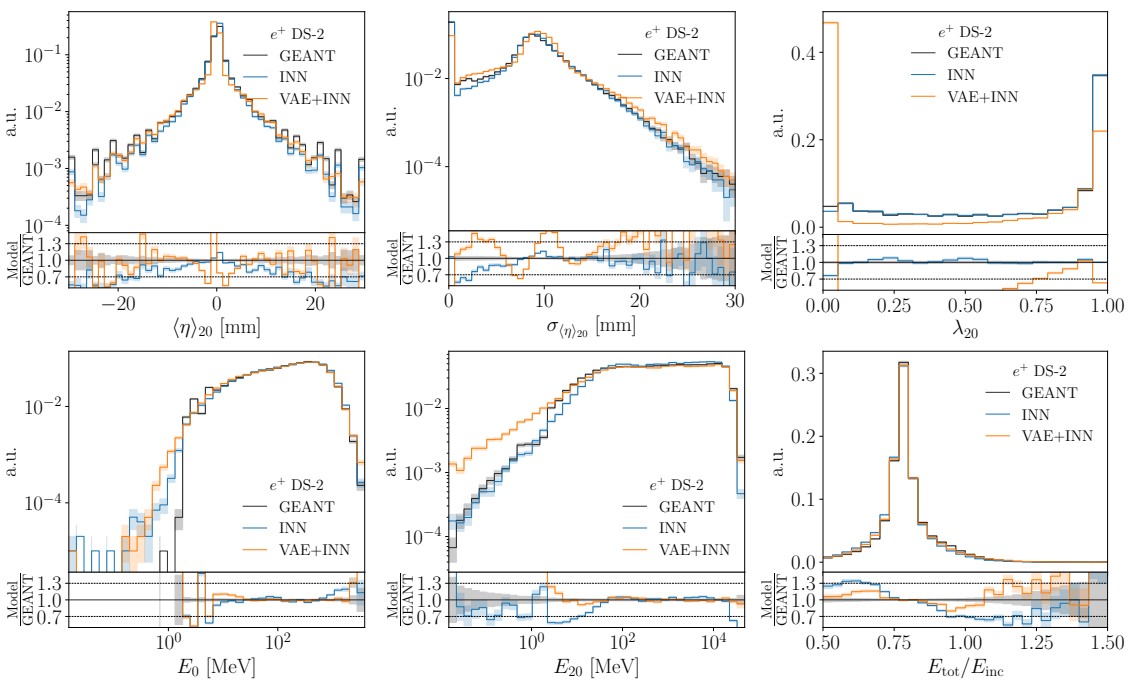

Figure 5: Set of high-level features for positron showers in dataset 2, compared between GEANT4, INN, and VAE+INN.

The classifier weights for the VAE+INN generating dataset 3 are shown in the right panel of Fig. 6. Even though the generative task is considerably harder, the learned weight distribution broadens centrally, but shows smaller tails than for dataset 2. The reason is that not only the generative network, but also our simple classifier are reaching their limits. However, the bulk

| | Batch size | INN | | | |
|---|---|---|---|---|---|
| | | 1-photon | 1-pion | 2-positron | 3-positron |
| GPU | 1 | $24.79 \pm 0.49$ | $24.76 \pm 0.35$ | $50.90 \pm 0.37$ | |
| | 100 | $0.385 \pm 0.002$ | $0.406 \pm 0.003$ | $1.900 \pm 0.026$ | |
| | 10000 | $0.162 \pm 0.002$ | $0.191 \pm 0.006$ | exceeding memory | |
| CPU | 1 | $17.48 \pm 0.09$ | $18.88 \pm 0.33$ | $117.5 \pm 1.8$ | |
| | 100 | $0.827 \pm 0.028$ | $1.004 \pm 0.047$ | $14.26 \pm 0.18$ | |
| | 10000 | $0.510 \pm 0.008$ | $0.719 \pm 0.016$ | $15.24 \pm 1.36$ | |
| | Batch size | VAE+INN | | | |
| | | 1-photon | 1-pion | 2 | 3 |
| GPU | 1 | $33.64 \pm 0.32$ | $33.54 \pm 0.23$ | $40.55 \pm 0.40$ | $43.13 \pm 1.4^*$ |
| | 100 | $0.507 \pm 0.005$ | $0.544 \pm 0.007$ | $1.05 \pm 0.02$ | $3.44 \pm 0.04$ |
| | 10000 | $0.180 \pm 0.002$ | $0.228 \pm 0.003$ | $0.748 \pm 0.018$ | — |
| CPU | 1 | $20.83 \pm 0.72$ | $20.05 \pm 0.13$ | $28.11 \pm 0.15$ | $39.46 \pm 1.1^*$ |
| | 100 | $0.582 \pm 0.005$ | $0.886 \pm 0.015$ | $1.94 \pm 0.01$ | $4.91 \pm 0.01$ |
| | 10000 | $0.328 \pm 0.004$ | $0.426 \pm 0.014$ | $1.25 \pm 0.01$ | $4.97 \pm 0.08$ |

Table 2: Per-shower generation times in ms. We show mean and standard deviation of 10 independent runs. The star indicates that only 10k samples were generated. The CPU timings were done with an Intel(R) Core(TM) i9-7900X at 3.30 GHz, the GPU timings with an NVIDIA TITAN V with 12GB RAM.

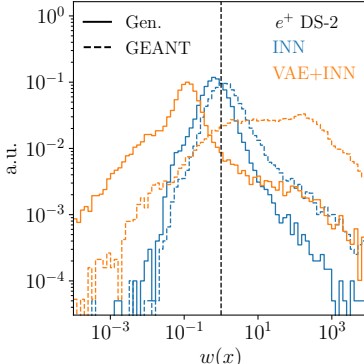 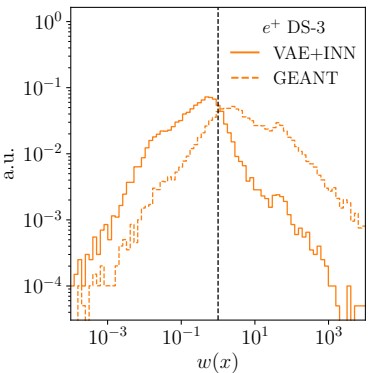

Figure 6: Classifier weight distributions. Classifier trained on $e^+$ showers on dataset 2 (left) and dataset 3 (right). The tails of dataset 3 should be taken with a grain of salt, giving the limitations of the simple classifier architecture.

of the classifier weight distribution clearly indicates that for dataset 3 the phase space density is mis-modelled by factors as large as 100 or 0.01 over large phase space regions. While the INN description of these position showers fails altogether, the VAE+INN results do not guarantee the level of precision we would expect for generative networks at the LHC.

## 4.5   Timing comparison

Finally, we time our networks using the CaloChallenge procedure. The INN architecture with modern coupling layers is ideally suited for fast and precise generation. We create a singularity container [112] of the software environment and take the time it takes to load the container, load the network, move it on the GPU, generate the samples, and save them to disk. In Tab. 2 we show the averaged results from ten runs. We observe a speed–up for increased batch size and when running on the GPU. The INN has a small advantage for dataset 1, but is unable to generate dataset 2 with the highest batch size and dataset 3 altogether. The VAE shows generation times at or below the millisecond mark.

## 5   Conclusions

Simulations are at the heart of the LHC program. Modern generative networks are showing great promise to improve their quality and speed, allowing them to meet the requirements of the high-luminosity LHC. In this paper, we have studied fast and precise normalizing flows, specifically an INN and a VAE+INN combination to generate calorimeter showers in high-dimensional phase spaces. As reference datasets we use the CaloChallenge datasets 1 to 3, with an increasing number of 368, 533, 6480, and 40,500 voxels.

For the simplest case, the dataset 1 photons, we have found that the INN generated high-fidelity showers and learns the phase space density of high-level features at the 10% level, except for failure modes which we can identify using high-level features and classifier weights over the low-level phase space. For this dataset, the VAE+INN shows no advantage, but less expressivity for example affecting the sparsity. For the pions in dataset 1 the INN faces more serious challenges, including mis-modeled features, and a wider range of learned classifier weights. The performance of the INN and the VAE+INN becomes much more similar.

For the positron showers in dataset 2 we have observed distinct advantages of the two networks, indicating that the compression by the VAE+INN helps learning the main features, but causes problems with low-energy depositions and the sparsity. Finally, the positrons in

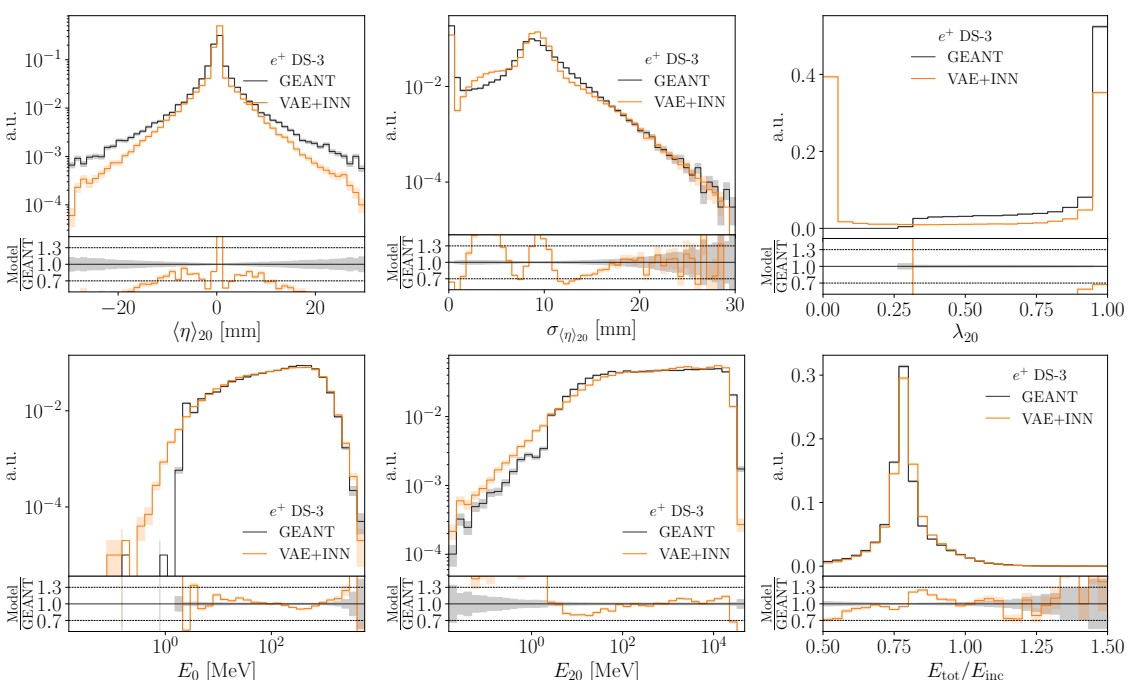

Figure 7: Set of high-level features for positron showers in dataset 3, compared between GEANT4, INN, and VAE+INN.

dataset 3 exceed the power of the plain INN, leaving us with the VAE+INN as the remaining option. The problem is that for this kind of phase space dimensionality the notion of precision-density estimation over phase space might not hold anymore for our network architectures.

## Acknowledgements

We would like to thank Theo Heimel, Stefan Radev and Peter Loch for helpful discussions. We would like to thank Thorsten Buss for collaborating in an early phase of the project. We would like to thank the Baden-Württemberg Stiftung for financing through the program *Internationale Spitzenforschung,* project *Uncertainties – Teaching AI its Limits* (BWST_ISF2020-010). DS is supported by the U.S. Department of Energy under Award Number DOE-SC0010008. This research is supported by the Deutsche Forschungsgemeinschaft (DFG, German Research Foundation) under grant 396021762 – TRR 257: *Particle Physics Phenomenology after the Higgs Discovery* and through Germany's Excellence Strategy EXC 2181/1 – 390900948 (the *Heidelberg STRUCTURES Excellence Cluster*).

# A   Network details

In this appendix we give some details on the network architectures and the preprocessing. The INN and the VAE+INN take layers normalized by the layer energy as input. The extra energy dimensions, calculated as in Eq. 4, are appended to the feature vector.

In the INN, we apply uniform noise and and a regularized logarithmic transformation with strength $\alpha$. The transformation applied to the features is a rational quadratic spline [108] for dataset 1 and a cubic spline [109] for dataset 2. The prediction of the spline parameters is obtained with an MLP sub-network with 256 nodes for each hidden layer. To equally learn each dimension, we permute the order of the features after a transformation and normalize the output to mean zero and unit standard deviation with an ActNorm [103] layer. In the large-scale architecture, we stack twelve blocks to construct the INN with the additional preprocessing block.

The VAE preprocessing has a similar structure. After normalization, we apply an $\alpha$-regularized logit transformation and a normalization to zero mean and unit standard deviation to each feature. We do not add noise during training and we set the latent dimension to 50 for dataset 1 and 2, and to 300 for dataset 3. We provide the full list of parameters in Tabs. 3 and 4.

| Parameter | INN DS1/DS2 | INN (with VAE) |
|---|---|---|
| coupling blocks | RQS / Cubic | RQS |
| # layers | 4 / 3 | 3 |
| hidden dimension | 256 | 32 |
| # of bins | 10 | 10 |
| # of blocks | 12/14 | 18 |
| # of epochs | 450 / 200 | 200 |
| batch size | 512 / 256 | 256 |
| lr scheduler | one cycle | one cycle |
| max. lr | $1 \cdot 10^{-4}$ | $1 \cdot 10^{-4}$ |
| $\beta_{1,2}$ (ADAM) | $(0.9, 0.999)$ | $(0.9, 0.999)$ |
| $b$ | $5 \cdot 10^{-6}$ | / |
| $\alpha$ | $1 \cdot 10^{-8}$ | $1 \cdot 10^{-6}$ |

Table 3: Network and training parameters for the pure INN.

| Parameter | VAE | |
|---|---|---|
| lr scheduler | Constant LR | |
| lr | $1 \cdot 10^{-4}$ | |
| hidden dimension | 5000, 1000, 500 (Set 1) | |
|  | 1500, 1000, 500 (Set 2) | |
|  | 2000, 1000, 500 (Set 3) | |
| latent dimension | 50 (Set 1,2) / 300 (Set 3) | Inner VAE |
| # of epochs | 1000 | |
| batch size | 256 | |
| $\beta$ | $1 \cdot 10^{-9}$ | |
| threshold $t$ [keV] | 2 (Set 1) / 15.15 (Set 2,3) | |
| hidden dimension | 1500, 800, 300 | |
| kernel size | 7 | Kernel |
| kernel stride | 3 (Set 2), 5 (Set 3) | |

Table 4: Network and training parameters for the VAE-INN.

The classifiers trained for the evaluation of the generative networks are simple MLP net-

works with leaky ReLU. We use three layers with 512 nodes each and a batch size of 1000. The network is trained for 200 epochs with a learning rate of $2 \cdot 10^{-4}$ and the Adam optimizer with standard parameters. To prevent overfitting, especially for the larger datasets, we apply 30% dropout to each layer, and we reduce the learning rate on plateau with a decay factor of 0.1 and decay patience of 10. The splitting between training, validation, and testing is 60/20/20%. The selection of the best network is based on the best validation loss.

## B  CaloGAN dataset

In this section we discuss the INN performance on the even simpler CaloGAN dataset [32, 34]. The INN architecture is described in Sec. 3. To extract uncertainties from the generative network, we promote the deterministic INN to its Bayesian counterpart [67, 113]. The implementation follows the variational approximation substituting the linear layer with a mixture of uncorrelated Gaussians with learnable means and a diagonal covariance matrix. In practice, we only upgrade the last layer of each sub-network to a Bayesian layer [114].

Figure 8 showcases two high–level features as examples of the performance of the CaloINN as compared to the training data distribution generated by GEANT4. We show the brightest voxel distribution in layer 0, the average $\phi$ location of the showers in layer 2, and the width of the shower depth width defined as the standard deviation of $s_d$ [41], with

$$s_d = \frac{\sum_{k=0}^{2} k E_k}{\sum_{k=0}^{2} E_k}. \tag{10}$$

The error bars in the GEANT4 distribution are the statistical errors while for the INN we estimate the uncertainties by sampling $N = 50$ times from the network and resampling the network parameters each time.

To evaluate our model on low-level observables, we resort again to classifier-based metrics. As already studied in a previous work [73], the INN samples are indistinguishable from the GEANT4 counterpart besides a few specific phase-space regions. We train a classifier on the

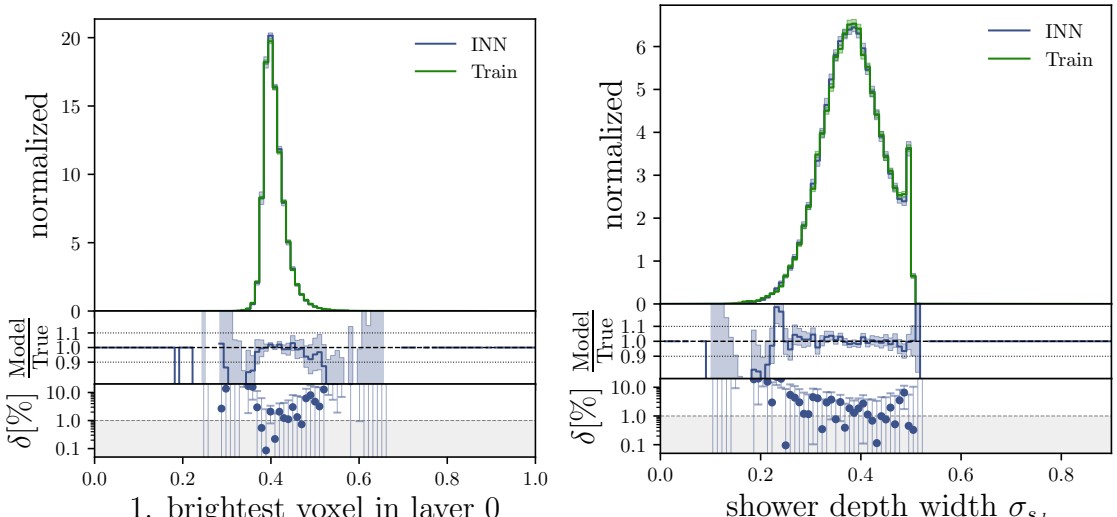

Figure 8: Comparison between CaloINN and GEANT4 on three high level features. Brightest voxel distribution in layer-0 (left), $\phi$ coordinate of the center of the shower in layer-2 (right), and width of the shower depth (bottom). Error bars on the INN are calculated after sampling from the Bayesian network $N = 50$ times.

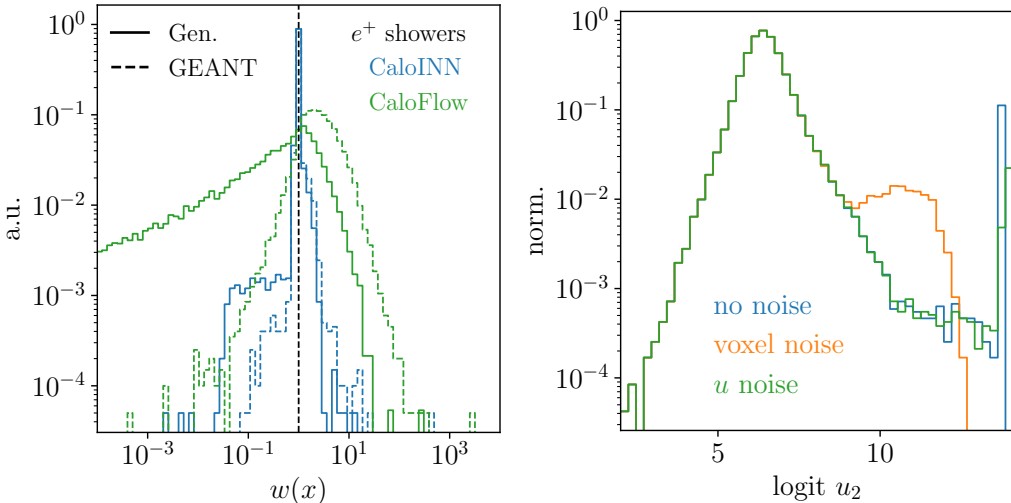

Figure 9: (left) Weight distribution of CaloFlow and CaloINN for $e^+$ showers. (right) $u_2$ distribution with different noise injections.

CaloFlow samples and find a large tail towards small weights. From clustering of the tail, we observe a clear dependence on the energy deposition total energy deposition. We link this effect to the learned energy variable $u_2 = E_1/(E_1 + E_2)$ and the noise injection procedure. If the noise is added at voxel-level, before calculating the additional energy variables, the flow learns distorted energy ratio distributions. Especially in the last layer, where the average energy deposition is smaller, this effect is larger. We summarize this effect in Fig. 9. We also provide the AUCs and the generation timings in Tab. 5.

| AUC | | CaloFlow [41] | CaloINN |
|---|---|---|---|
| $e^+$ | unnorm. | 0.859(10) | 0.525(2) |
| | norm. | 0.870(2) | 0.598(3) |
| | hlf | 0.795(1) | 0.656(2) |
| $\gamma$ | unnorm. | 0.756(50) | 0.530(2) |
| | norm. | 0.796(2) | 0.584(2) |
| | hlf | 0.727(2) | 0.671(2) |
| $\pi^+$ | unnorm. | 0.649(3) | 0.662(2) |
| | norm. | 0.755(3) | 0.735(4) |
| | hlf | 0.888(1) | 0.786(4) |

| | Batch size | CaloFlow [43] | CaloINN |
|---|---|---|---|
| GPU | 1 | $55.12 \pm 0.19^*$ | $23.79 \pm 0.10^*$ |
| | 100 | $0.744 \pm 0.04$ | $0.425 \pm 0.005$ |
| | 10000 | $0.249 \pm 0.003$ | $0.211 \pm 0.003$ |
| CPU | 1 | $119.9 \pm 0.9^*$ | $46.39 \pm 3.18^*$ |
| | 100 | $3.13 \pm 0.11$ | $1.14 \pm 0.03$ |
| | 10000 | $1.681 \pm 0.004$ | $0.72 \pm 0.01$ |

Table 5: (left) AUC of the two classifiers trained on the CaloFlow teacher and CaloINN samples. (right) Per shower generation timings in ms. We show mean and standard deviation of 10 independent runs of generating 100k showers. The star indicates that only 10k samples were generated in total.

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
