# Peer review of "Normalizing Flows for High-Dimensional Detector Simulations"

_SciPost Physics_

## Round 1 · Referee Report · Anonymous (Referee 1) · 2024-6-17

Report

Generative models are a prime candidate to speed up full detector simulations in HEP. A lot of previous work is referenced in this paper. The so-called CalChallenge community benchmark is used featuring different levels of difficulty in the simulation task / dimensionality.

Two solutions are presented: 1) flows based on coupling layers and 2) similar kind of flows but in a learned latent space of a VAE with reduced dimensionality to cope with high-dimensional data

Main comments:

Network parameters are given in the appendix but little intuition is provided as to the choice of hyper parameters. No mention of an ablation study is given. A justification of parameter choices is desirable. In particular in light of the comparisons of the two models. The reader wonders if the different performance is really due to the architecture or due to non-optimal parameter choices.

The low-level classifier is in principle a useful one-catch-it-all metric. However, it comes with downsides of non-reproducibility and a hard to interpret metric. Is an AUC of 0.601 good enough? And is 0.805? (Also suggest to round to significant digits) The reader has no way of judging the quality of the surrogate model based on these numbers. A discussion needs to be offered to the reader.

Minor comment:

The reader wonders what is meant by the last sentence: “the notion of precision-density estimation over phase space might not hold anymore for our network architectures.” A reformulation is in order.

Recommendation

Ask for minor revision

---

## Round 1 · Referee Report · Anonymous (Referee 2) · 2024-6-20

Strengths

1 Innovative models 2 All datasets from the Calorimeter Challenge are used giving a full picture of the capabilities of the tools 3 Extremely high performance on the photon dataset

Weaknesses

1 Lack of comparison with other models
2 Need to have a more granular evaluation of results
3 Missing future directions and lesson learned

Report

I reviewed the paper titled "Normalizing Flows for High-Dimensional Detector Simulations" by F. Ernst, L. Favaro, C. Krause, T. Plehn and D. Shih. The work was carried out as part of the Fast Calorimeter Simulation Challenge, all three datasets were considered, thus providing a comprehensive picture of the tool's performance.

This paper describe two novel tools to generate calorimeter showers based on invertible neural networks (INN) and the combination of INN with a VAE. INNs are needed for producing a fully invertible chain from generation to data analysis, enhancing the interpretability of data collected by HEP experiments. I find that the paper meets the criteria for publication, given the extremely high performance of the INN on the Calorimeter Challenge datasets.

The paper is well written with only a few paragraph that should be improved for better readability, particularly the architecture section.
The work can be re-derive using the provided code (though it should be tagged) and the training and evaluation datasets are publicly available.
The citations are sufficient but some need updating as the preprints have been published.
The conclusion needs to be expanded to include comparison with other models to place this tool correctly in the landscape of available tools.
A section on future directions, lesson learned and possible improvement should be added.
The abstract would benefit from some expansion, mentioning results, while the introduction is in good shape with only minor comments affecting it.
It is my opinion that this paper is worth publishing once all my concerns are addressed.

My main concern is the lack of in-depth evaluation of the two models. A more granular comparison should be made for different energies and layers to provide a better insights into the tools' performance, limitations, and deployment potential.
As already mentioned, the paper does not place itself on the map by comparing with published models. Adding such section would significantly strengthen the paper.

I hope my comments will be taken as an encouragement to improve the quality of your work which I consider extremely high.

Kind regards.

Abstract
This is a rather underwhelming abstract, please consider mentioning some of the results you achieved.

Introduction
"... is the advent of deep generative networks." Probably better to change to use the word "is the advent of generative models". Change the next sentence accordingly

I would also like to remind the authors that fast simulation existed well before the advent of deep learning, most ATLAS measurements in Run2 used AtlFast2 fast simulation which is based on parametric models. Indeed AtlFast3 is still using refined parametric model to a large extent. Please mention this and do not present fast simulation as something invented thanks to ML.

"... have entered LHC physics". Please tone down this statement, no measurement in LHC is using any of the work referenced. These are all proof-of-concept papers that have not been applied to real LHC physics (you know, systematics studies and a scrupulous review from a 3000+ collaboration). Making these proposals into production or even into a paper is a multi year major endeavour (which is why it is rarely done).

"... gains in both speed and accuracy ...". While this may be true, it is not evident from the result section in the paper, no comparison is provided to assess a gain in either speed (in what? training, inference or both?) or accuracy. Please add some comparison as suggested in the comments for the conclusion section (but feel free to place where you prefer).

Dataset
You should probably change the first sentence which is almost an exact repetition of a sentence half page above. Actually, changing the first one in the introduction is probably a better idea; you could use something like "As stated above, the datasets used in this work were provided as part of the CaloChallenge and represent three increasing dimensionalities from the current LHC calorimeter granularity to the ultra high granularity of future calorimeters proposed for ILC, CLIC, FCC and beyond." (with all the relevant references). This is more high level and provides further motivation for your work.
"Einc = 256 MeV ... 4.2 TeV" please not that the value you quote is the momentum of the particle that was generated. It is the energy only for the photons, for 256MeV pions the difference is significant given the mass of the pion. Actually, the energy that should be used for interpolating the samples is the kinetic energy of the particle, depending on how the conditioning is done, this may not be a crucial point, but it is relevant for using the tool for actual simulation where the interpolation will be used. Please consider updating your pion results accordingly, you may have better performance in the low energy samples.

CaloINN

"We replace the standard affine layer by a more expressive spline transformation."
Can you elaborate on this sentence and the following ones? Why is this needed? How did you optimise this change? Can you relate number of parameters to number of voxels or other parameters in the datasets? As it stand, the choice is rather arbitrary and need more precise motivations.

"The problem with the INN is the scaling towards dataset 3 with its high-dimensional phase space of 40k voxels."
As above, can you be more specific in what constitute a limit for the INN of 3.1? Is is memory, time or something else?

"We provide the code in a Github repository"
Can you have a tag for the version of the code used for the paper?

"...because it outperforms for example its Gaus?sian counterpart" --> " because it outperforms other methods/functions/approaches. Fro example, the Gaussian... Similarly, the continuous ... The best solution is the Bernoulli decoder defined as..."

"Applying a 2-step training" add a comma

General comment: The description is rather complex to follow, please consider improving the flow by itemising the steps or, even better, providing some visual help (a figure with a scheme).

"Our assumption is that the calorimeter layers do not require a full correlation"
This is good approximation for dataset 2 and 3 because of the nature of EM showers, where a layer by layer correlation is enough. You could use a similar assumption for photons in dataset 1 but not for pions. Please add a physics motivation to the text to justify your assumption.

Results
"since photons only undergo a handful interactions in the calorimeter". This sentence is not clear, I assume you mean that only pair production (gamma->e+e-) and Bremsstrahlung are relevant processes for the energy considered. Please consider expanding the text.

"In this established benchmark normalizing flows" --> "In this established benchmark, normalizing flows" (add a comma)

"For instance for the calorimeter layer 2". I think you mean layer 1 here. This is the asymmetric layer where the centre of mass along lambda is shifted by 1 mm forward.
Figure 1 also shows the label "2" for the first two figures, while you probably want to show layer 1

General comment for dataset 1: Please consider showing plots for different energy points rather than having inclusive plots. Inclusive plots have very little physical meaning as you will be dominated by different energies in different regions. You have done this for the total energy and it is clear from Fig2 that only the two lowest energy points are the reason for the mismodelling of the bottom left plot in Fig 1. Similarly, the energy in each layer is different depending on the energy. The E2 is plot is dominated by the highest energies, so you cannot assess well the mid and low energy points. You do not have to present all energy points for every plot, 3 incident energies will be enough to explain which energies are well or poorly modelled for the specific observable.
Please add total energy in layer 1 for dataset 1 photons as this is a layer with an important energy deposit (it is the majority below 4 GeV).
Please add plots for the 5th and 6th layers (layers 12 and 13) for energy and shapes for dataset 1 pions. These are the two main hadronic layers that have most of the energy from the showers.
Add more comments and description using these figures.
The differences observed in the classifier weights may be explained by specific difference that may or may not have a significant physics impact. For example, a mismodelling of the 1-4 TeV pions will have a very limited impact as these are extremely rare in jets. On the other hand, small deviation in pions of 10-50 GeV can result in significant physics limitation as these are the most common energies for pions in jets.

"and that adding a VAE to compress the information does not help with the network performance" and in other places
This is not accurate, the VAE actually degrade the performance of your INN, so it is not neutral as in your statement ("does not help") but it has a detrimental effect. Please rephrase here and in DS2 and in conclusion.

General comment for dataset 2 and 3: As for dataset 1, please add more figures for a few more layer and different energy ranges. Three ranges can be enough (1-10, 10-100, 100-1000)
Similarly to the photons in dataset 1, at low energy the deep layers will not be reached by the showers. Dividing in different energy ranges will provide a better insight of the performance of the tool.

General comment on results: why do you only present low level comparison? High level is also important if not more so than the low level that may be easily be below the threshold for noise or clustering. Please consider adding it as it should not be much work as you already have the low level comparison.
Can you provide some motivation for the different performance between photons and the electrons in DS2? in principle it is the same electromagnetic shower, simply sampled with a higher granularity. Could you make your networks bigger to account for it this difference or have a different pre- and post- processing where you reduce the dimensionality by first merging voxels and then splitting again the energy after generation? This does not need to be a new fully fledged study, but something to consider for the "future direction" part I suggest below.
Time comparison
Can you also add the training time? This is a relevant parameter as the datasets cover only a small slice of the detector; extrapolating to the full detector is problematic given the difference in geometry that normally requires different a voxelisation. Ultimately, multiple training will be required for different eta and particles.
The training time is also an interesting parameter as it can be a limiting factor in optimising a model.

Conclusion
This section is a bit short and should also give a more general overview of your results and relate them to the state-of-the-art in the field, for example comparing them to other models. You can use any figure of merit your like, there are plenty to pick and use for comparisons.
For example, the INN for photons is one of the best model on the market, if not the best. This does not stand out

A (sub)section on future work, possible improvements and prospect should be added too. It would be great to see some comments on solutions that you tried but did not work; science is as much about success as it is about failures (so others do not have to make them).

Appendix B
I suggest to use layer 1 or 2 for your example plot. Layer 0 is not as relevant as layer 1 or 2 for the amount of energy deposited and for particle ID.

References
I have not checked every single reference but there are a few that are now published, specifically 48, 55 and 56. Please check all references and make sure to correctly quote them.

Requested changes

1 Add a section or expand the conclusion adding a comparison with other models
2 Expand the result section with more in-depth evaluation of the models for different energies and layers
3 Improve text as suggested

Recommendation

Ask for minor revision

---

## Round 1 · Referee Report · Anonymous (Referee 3) · 2024-7-5

# Referee Report on "Normalizing Flows for High-Dimensional Detector Simulations"

The paper "Normalizing Flows for High-Dimensional Detector Simulations" investigates the performance of normalizing flows for fast calorimeter shower simulations with increasing phase space dimension. The authors present benchmarks for invertible networks applied to the CaloChallenge datasets and introduce a VAE+INN approach to address scaling issues for higher-dimensional phase spaces.

The paper is well-structured and provides a thorough analysis of normalizing flows for detector simulations. The benchmarks and comparisons are clearly presented, and the introduction of a VAE to address high-dimensional phase spaces is a valuable contribution.

The combination of VAEs with invertible neural networks to process high-dimensional data is innovative and represents a major challenge in this field. Furthermore, the authors provide a detailed evaluation of the proposed methods using high-level features and classifier weights. Finally, the results are presented clearly and concisely so that it is easy to track the performance of the different approaches. Making the code available in a GitHub repository promotes transparency and reproducibility, which is highly commendable.

Overall, the work makes a significant contribution to the field of high-dimensional detector simulations using normalizing flows. The methods are sound, and the results are promising. With minor revisions, I recommend the paper for publication in SciPost Physics after considering the comments below.

- The paper lacks a detailed discussion of the hyperparameter selection process in all models used in this work. The inclusion of this information would help to understand the tuning process and its impact on the results.

- The paper would benefit from a quantitative comparison with other existing methods using specific metrics. This would allow for clearer benchmarking and highlight the relative strengths and weaknesses of the proposed methods compared to other methods in the literature.

- The authors use of a very small beta value (e.g. $10^{-9}$) in the loss function in addition to the binary cross entropy. It would be helpful if the authors could explain how this choice benefit the results and why is so.

- While the paper contains several references to normalizing flows for detector simulations, it could benefit from citing other recent work that has made important contributions in this area. Furthermore, there are no references to the $\beta$-VAE model, which is used extensively in the literature.

---

## Round 2 · Referee Report · Anonymous (Referee 1) · 2024-12-9

Report

The authors have addressed all my questions and comments satisfactorily. I find the manuscript suitable for publication in SciPost and recommend its acceptance.

Recommendation

Publish (meets expectations and criteria for this Journal)

---

## Round 2 · Referee Report · Anonymous (Referee 3) · 2024-12-23

Report

All my comments were addressed

Recommendation

Publish (meets expectations and criteria for this Journal)

---

## Round 2 · Referee Report · Anonymous (Referee 2) · 2024-12-23

Report

All my comments are addressed and I am happy to proceed with publication.

Recommendation

Publish (meets expectations and criteria for this Journal)

---

## Round 2 · Author Response

We would like to thank the referees for their reports. In particular, we appreciated the detailed report from Referee 2. Our aim for a discussion of meaningful features in calorimeter showers, rather than presenting a large list of figures, greatly benefit from their comments.
We additionally publish the sample in a Zenodo repository with the full set of high-level observables studied.
In light of the extremely encouraging reports from the referees and the promising results, we believe our manuscript meets the criteria for the publication in Scipost Physics.

Main comments:
Regarding the main comments from the referees, section 4.5 now contains a comparison to similar normalizing flows and to a diffusion model in terms of shower generation time and accuracy.
We added a timing comparison and the classifier AUC for high- and low-level features.
We did not perform and in-depth ablation study because of resource constraints. We expanded the appendices with a discussion on the hyperparameters selection and the small grid search done.

---

## Round 2 · List of Changes

Report 1
See main comments.

Report 2

We adopted almost all the text improvements suggested by the referee. We kept the usage of "networks" when explicitly referring to a model defined by a neural network.

Abstract
We extended the abstract with the bottom line of our results.

Introduction
We added a paragraph on the effort done in ATLAS and CMS on fast detector simulations.
The claim is now supported by Sec.4.5 and the appendix on the CaloGAN dataset.

Dataset
We adopted all the suggested changes.

CaloINN
We expanded the text describing the INN, including the scaling aspect with input dimensionality. We also include a new schematic representation of the INN workflow.
We tagged the current state of the Github repo, which only contains refactorizations of the code at the time of submission.

VAE+INN
We added a VAE+INN flowchart and the motivation for a kernel-based encoding/decoding step.

Results
We clarified the first paragraph on photon showers.
We moved the inclusive set of high-level observables to the appendices. As suggested, we show and discuss three incident energies. The layers are chosen such that we discuss where there is the largest energy deposition.
We adopt the same scheme for the pion showers.

For dataset 2, we divided the high-level feature figures in the three E_inc windows (1-10), (10-100), (100-1000) GeV and selected only one layer to discuss the networks.
To avoid overcrowding the paper with figures, we have not done the same for dataset 3 but we included all the histograms in the published Zenodo repository.

We added a timing comparison in Table 3, the high-level classifier in Table 4, and the approximate training time on our cluster in the text.

We clarified the comparison between the INN and the VAE+INN throughout the text.

Conclusion
In the conclusions we expanded the discussion of the performance of our networks on all the studied metrics.
We comment on the observed differences between photon showers in dataset 1 and the electrons in dataset 2.
We give an outlook of what can be improved and directions for future works.

We updated all the references.

Report 3

VAE+INN
1-2 See main comment

3 We expanded the discussion on the selection of the beta parameter. The small value used ensures that the latent space is compact and therefore learnable by the latent INN.

4 We added citations to the beta-VAE and other latent models.

---

## Editorial Decision

accepted_in_target_journal